# Kinetics of Whey Protein Glycation Using Dextran and the Dry-Heating Method

**DOI:** 10.3390/foods8110528

**Published:** 2019-10-25

**Authors:** Na Li, Abhiram Arunkumar, Mark R. Etzel

**Affiliations:** 1Department of Food Science, University of Wisconsin, 1605 Linden Drive, Madison, WI 53706, USA; nli45@wisc.edu; 2Voyager Therapeutics, 75 Sidney St., Cambridge, MA 02139, USA; arunkumar@uwalumni.com

**Keywords:** conjugation, dairy, carbohydrates

## Abstract

Glycation of proteins by polysaccharides via the Maillard reaction improves the functional properties of proteins in foods, such as solubility, heat stability, emulsification, foaming, and gelation. Glycation is achieved by either the dry heating or the wet heating method, and considerable research has been reported on the functionality of the reaction mixture as tested in foods. While the characteristics of the glycates in foods have been well studied, the kinetics and equilibrium yield of the protein-polysaccharide glycation reaction has received little attention. Industrial manufacture of the glycates will require understanding the kinetics and yield of the glycation reaction. This work examined the glycation of whey protein isolate (WPI) and glycomacropeptide (GMP) by using dextran and the dry-heating method at 70 °C and 80% relative humidity. The disappearance of un-glycated protein and the creation of glycated protein were observed using chromatographic analysis and fluorescence laser densitometry of sodium dodecyl sulfate-polyacrylamide gels. Data were fit using a first-order reversible kinetic model. The rate constants measured for the disappearance of un-glycated protein by sodium dodecyl sulfate-polyacrylamide (SDS-PAGE) (k = 0.33 h^−1^) and by chromatographic analysis (k = 0.38 h^−1^) were not statistically different from each other for WPI-dextran glycation. Dextran glycation of GMP was slower than for WPI (k = 0.13 h^−1^). The slower rate of glycation of GMP was attributed to the 50% lower Lys content of GMP compared to WPI. Yield for the dry-heating dextran glycation method was 89% for WPI and 87% for GMP. The present work is useful to the food industry to expand the use of glycated proteins in creating new food products.

## 1. Introduction

Glycation of proteins by the Maillard reaction is a promising food-grade method to modify food proteins [1,2,3,4,5,6]. Glycated proteins have improved functional properties in foods such as superior solubility, heat stability, emulsification, foaming, and gelation properties [3,7]. Glycation of proteins may help people who suffer from food protein allergies by lowering IgE-binding capacity [8]. Glycation uses the first step in the Maillard reaction to create a reversible Schiff-base linkage between the free amino moiety in the protein and the carbonyl moiety in the polysaccharide [9,10,11,12,13]. The Schiff-base linkage can be created via the dry-heating method [9,14] or the wet-heating method [15,16]. In the dry-heating method, an aqueous mixture of the protein and polysaccharide is first dried and then heated for a certain time (2 h to 9 days) at a fixed temperature (60 to 130 °C) and relative humidity (60 to 80%). In the wet-heating method, the aqueous solution is heated for a specific time (2 h to 2 days) at a fixed temperature (60 to 95 °C).

Schiff base formation is a reversible condensation reaction that generates water as a by-product. By Le Chatelier’s principle [17], the presence of water in the reaction mixture drives the reaction in reverse, lowering yield. For example, the yield of glycation using the wet method is low (<5%) [15,16]. Subsequently yield was increased to 18% using a reaction time of 24 h at 60 °C [18]. The dry-heating method uses a desiccator to remove water generated by the condensation reaction, shifting the chemical equilibrium towards product formation, which increases yield. For example, whey protein-maltodextrin powders heated for 2 h at 80 °C in a desiccator at 79% relative humidity formed substantial amounts of glycates with little un-glycated protein remaining in the reaction mixture [14].

The purpose of the present study was to measure the time course and equilibrium yield of the glycation reaction between dextran and whey protein isolate (WPI) or glycomacropeptide (GMP) using the dry-heating method. The reaction between polysaccharides and proteins via the Maillard reaction is different from the reaction between simple sugars and proteins. In either case, the Maillard reaction pathway starts with the formation of a Schiff base between the free amino moiety in the protein and the carbonyl moiety in the carbohydrate [19]. For simple sugars the Schiff base undergoes spontaneous rearrangement to either an Amadori or Heyn’s compound [20]. Protein-polysaccharide glycation products are not subject to post-Amadori–Maillard reaction steps [4]. The Maillard reaction stops at Schiff base formation [15]. The protein-polysaccharide product is colorless and has no odor [4]. It was not the purpose of the present work to explore the sequence of elementary chemical reactions that make up the Maillard reaction pathway by using the tools of chemical kinetics. Rather, the present work was aimed at measuring the time course and equilibrium yield of the glycate formation reaction and extracting apparent rate constants using a simple mathematical model and a fitting procedure.

In order to determine the time course and equilibrium yield, a quantitative method was required to measure the concentration of reactants (un-glycated protein and dextran) and products (glycated protein) versus time. We used a new method for the chromatographic analysis of whey protein-dextran glycation products and a complementary fluorescence laser densitometry method for this purpose. The present work is important because producing glycates quickly and in high yield is essential for delivering on the potential benefits of this new food-grade method for protein modification.

## 2. Materials and Methods

Whey protein isolate (WPI) and glycomacropeptide (GMP) were from Davisco Foods International (Le Sueur, MN, USA). WPI contained 92.7% protein, 2.0% ash, 5.0% moisture, 0.0% lactose, and 0.3% lipids. GMP contained 86% protein, 6.5% ash, 6.0% moisture, 1.0% lactose, and 0.5% lipids. Dialysis for removal of the 1% lactose in GMP was deemed unnecessary based on past research [4]. Dextran T10 was from Pharmacosmos Company (Holbaek, Denmark) and had an average molecular mass of 5.2 kDa. Precast gels (Tris-Glycine, 4–20% linear gradient, 18 wells), pre-stained molecular mass standards, Tris/glycine/SDS premixed buffer, Laemmli sample buffer, and Coomassie Blue G-250 stain were from Bio-Rad Laboratories (Hercules, CA, USA). Pierce GelCode glycoprotein staining kit was from Thermo Fisher Scientific (Waltham, MA, USA). SYPRO Red Protein Gel Stain was from Lonza Rockland (Rockland, ME, USA). Other chemicals were from Fisher Scientific (Pittsburgh, PA, USA). Buffers were prepared at 22 °C. Centrifugal filter units (Amicon Ultra-15, 3 kDa) were from MilliporeSigma (Burlington, MA, USA).

### 2.1. Synthesis of Glycated Proteins

Glycation was conducted using the dry-heating method. Dextran and WPI or GMP were dissolved in 10 mM sodium phosphate buffer (pH 6.5) at a 3:1 mass ratio, resulting in a dextran-WPI molar ratio of 10:1. The liquid reaction mixture was frozen, lyophilized and ground into a powder using a mortar and pestle to form particles of about 0.1–3 mm diameter, and then dispensed into seven 20 mL glass scintillation vials. All seven vials without caps were equilibrated to 85% relative humidity at 22 °C for 24 h in a desiccator containing saturated potassium chloride solution. One vial was taken before the start of the reaction and capped. The remaining six un-capped vials were placed into a pre-heated desiccator at 70 °C that contained saturated potassium chloride solution giving a relative humidity of 80%. At time points of 1, 2, 4, 8, 16, and 32 h, one vial was taken from the desiccator and capped. The seven vials were stored at −20 °C prior to analysis.

### 2.2. Gel Electrophoresis

Sodium dodecyl sulfate-polyacrylamide (SDS-PAGE) gel electrophoresis was performed to measure the progress of the glycation reaction, following the protocol of Bund et al. [18]. Dried reaction products were fully dissolved in water, subjected to the SDS sample preparation procedure, loaded into the gel, and visualized by Coomassie Blue staining. Precision Plus Protein Kaleidoscope Pre-stained Protein Standards (ST) were also applied on the Coomassie gel as molecular mass makers. The ST mixture contained ten recombinant proteins of molecular mass 10 to 250 kDa. A Pierce GelCode glycoprotein staining kit was used for the detection of glycoproteins in the glycoprotein stained gel.

Coomassie and glycoprotein stained gels provided a qualitative result, but more quantitative data were needed for the kinetic analysis. Therefore, fluorescence laser densitometry after staining by SYPRO Red was used for protein quantification. Gels were scanned on TYHOON FLA 9000 laser densitometer (GE Healthcare, Piscataway, NJ, USA) in fluorescence mode using an excitation at 532 nm and emission at 610 nm. Bands were quantified using ImageQuantTL software (GE healthcare). Calibration of band volume to protein concentration was accomplished by applying internal standards to three lanes of the gel (lanes A, B, C). The internal standards consisted of three liquids each containing known concentrations of ALA and BLG of 0.1 to 0.3 mg/mL such that the total was 0.4 mg/mL.

### 2.3. Chromatographic Analysis

Glycated samples of WPI were analyzed by cation exchange chromatography using a 5 mL HiTrap MacroCap SP column from GE healthcare (Marlborough, MA, USA) connected to an ÄKTA Explorer 100 HPLC system (GE Healthcare). Glycated protein samples were reconstituted in 50 mM sodium lactate, pH 4.0 (Buffer A) and syringe filtered using a 0.22 µm PVDF filter (MilliporeSigma, Burlington, MA, USA). The chromatographic method consisted of 4 steps: (1) column equilibration using 5 column volumes (CV) of buffer A; (2) loading the sample into the column using a 2 mL sample injection loop to bind the glycated and un-glycated proteins to the column; (3) elution of the glycated protein using 12 CV of 40% buffer B (1 M NaCl in Buffer A); and (4) elution of unreacted un-glycated protein using 2.5 CV of 100% Buffer B. The column was then re-equilibrated using a 5 CV of buffer A and cleaned using 0.1 M NaOH. Glycated protein eluted in the “low salt” peak (40% buffer B) and unreacted un-glycated protein eluted in the “high salt” peak (100% buffer B). Unicorn 5.0 software was used to set up the chromatographic method and calculate peak area at 280 nm (PA_280_). Protein (µg) ≈ 25 × PA_280_, where 25 is the flow rate (5 mL/min) times the path length correction (10 mm/2 mm) for the detector flow cell.

### 2.4. Kinetic Model of the Glycation Reaction

In Schiff base formation, free amino acids on the protein react reversibly with carbonyls on the polysaccharide to produce glycated protein as shown in Equation (1) [4,9,21]:
P + D ⇔ PD + W(1)
where P is un-glycated protein, D is dextran, PD is protein-dextran glycate, and W is water. Dextran and water were in ten-fold molar excess or more in the present work and can be considered to be constant. The kinetic equations are then shown in Equations (2) and (3) [22]:[P] = [P]_eq_ + ([P]_0_ − [P]_eq_) e^−kt^(2)
[PD] = [PD]_eq_ (1 − e^−kt^)(3)
where k is the apparent rate constant and [P] and [PD] are the concentrations of un-glycated protein and protein-dextran glycate, respectively, at time t. The value of [P] at time zero is [P]_0_. The values of [P] and [PD] at equilibrium (t → ∞) are [P]_eq_ and [PD]_eq_, respectively. In the present work, measured values of [P] and [PD] versus time were used to obtain the fitted parameter values [P]_eq_, [PD]_eq_, and k.

### 2.5. Statistical Analysis

Equations (2) and (3) were fitted to the experimental data for the time course of the reaction by nonlinear regression using the JMP Pro software, version 11 (SAS Institute, Gary, NC, USA) to obtain the fitted parameter values [P]_eq_, [PD]_eq_, and k. Results were expressed as mean ± standard error. Point-by-point comparisons between the data and the fitted equations were made by *t*-test using SAS studio 3.5 (SAS Institute, Gary, NC, USA). The *p*-value was reported.

## 3. Results

### 3.1. Kinetics of Whey Protein Isolate-Dextran Glycation Using SDS-PAGE Analysis.

Samples from reaction times of 0, 1, 2, 4, 8, 16, and 32 h were analyzed by SDS-PAGE and stained using Coomassie Brilliant Blue (Figure 1A) and glycoprotein stain (Figure 1B). The ten ST bands in Figure 1A were from 10 pre-stained protein standards (10 to 250 kDa). The single ST band in Figure 1B was from the glycoprotein horseradish peroxidase (44 kDa). At 0 h, the sample consisted mostly of un-glycated alpha-lactalbumin (ALA, 14.4 kDa) and beta-lactoglobulin (BLG, 18.4 kDa), the main proteins in bovine cheese whey [23]. Increasing the reaction time decreased the amount of un-glycated ALA and BLG, and simultaneously increased the amount of glycated protein that appeared as a smear band at higher molecular mass. The glycated protein smear band darkened in intensity and the midpoint of the smear band shifted to a higher molecular mass as reaction time increased. By 32 h of reaction time, the ALA and BLG bands were much smaller than at 0 h of reaction time, meaning that most of the ALA and BLG had been glycated by dextran, and some of the glycated protein was larger than the largest marker band of 250 kDa.

As shown in Figure 2, similarly prepared samples of different reaction times were separated by SDS-PAGE and stained by SYPRO Red. A gradual disappearance of un-glycated ALA and BLG, and a gradual appearance of the glycated protein smear band were observed, consistent with the results shown in Figure 1.

The SYPRO Red stained gel of Figure 2 was analyzed by fluorescence laser densitometry. Un-glycated protein was quantified by referencing lanes A, B, and C as internal standards of ALA and BLG of known concentrations to construct a calibration curve. Results are shown in Figure 3. Concentration is in units of grams of protein in the sample per liter of solution after dissolution of the reaction mixture in water but before subjecting the liquid to the SDS sample preparation procedure. Un-glycated protein (WPI) decreased as reaction time increased and reached completion in about 8 h. The data in Figure 3 were fit using the kinetic model of Equation (2). The fitted parameter values were k = 0.33 ± 0.06 h^−1^ and [P]_eq_ = 0.04 ± 0.03 g/L. There was no statistical difference between the data points and the fit using the kinetic model (*p* > 0.05). The extent of glycation (ε) was calculated from ε = 1 − [P]/[P]_0_ using the experimental data at each reaction time. At 8 h, ε = 0.89, or 89% of the un-glycated protein had disappeared from the sample.

### 3.2. The Effects of Reaction Time on Glycation Using Chromatographic Analysis.

The SYPRO Red staining and laser fluorescence scanning method could be used to measure the loss of un-glycated protein but not the creation of glycated protein because the smear band of the glycated protein was not quantifiable. Therefore, the chromatographic method was used as an independent check of the SDS-PAGE result for un-glycated protein and as the only way to measure the simultaneous creation of glycated protein.

Figure 4 shows peak areas obtained using the chromatographic method. Example chromatograms are shown in our previous publication [10]. Un-glycated protein decreased exponentially during the first 8 h of reaction time. At 8 h, ε = 0.88, or 88% of the un-glycated protein had disappeared from the sample. The high-salt peak area data for un-glycated protein were fit to the model of Equation (2) for un-glycated protein. The fitted parameter values were k = 0.38 ± 0.06 h^−1^ and [P]_eq_ 60 ± 30 mAU-min. There was no statistical difference between the data points and the fit using the kinetic model (*p* > 0.05). The rate constant (k) measured from the disappearance of un-glycated protein by chromatographic analysis was not statistically differently from the rate constant (k) measured by SDS-PAGE analysis (*p* > 0.05). In other words, by two independent analytical measures (SDS-PAGE and chromatography), the same rate of reaction was observed.

The low-salt peak area data for glycated protein were fit to the model of Equation (3). The fitted parameter values were k = 0.18 ± 0.03 h^−1^ and [PD]_eq_ 850 ± 40 mAU-min. There was no statistical difference between the data points and the fit using the kinetic model (*p* > 0.05) except for the data point at 4 h for which the difference was marginal (*p* = 0.043). Compared to the rate constant for the disappearance of un-glycated protein as measured by SDS-PAGE (k = 0.33 h^−1^) and by chromatographic analysis (k = 0.38 h^−1^), the rate constant for creation of glycated protein (k = 0.18 h^−1^) was about half the expected value.

### 3.3. Kinetics of Glycomacropeptide-Dextran Glycation

The next subject of investigation was the glycation of GMP by dextran under the same reaction conditions as the glycation of WPI by dextran. Figure 5 shows the glycoprotein stain of the SDS-PAGE analysis of GMP-dextran samples taken at each reaction time. Samples taken at 0 h and 1 h show the characteristic “king’s crown” shape of unreacted GMP [24], and no glycation products from the dextran glycation reaction. GMP is naturally glycosylated by mucin-type carbohydrate chains at seven different Thr residues [25] and stains by glycoprotein stain. At reaction times of 2, 4, and 8 h, glycated GMP appears as seen by the increasingly larger smeared bands as time increases, and the disappearance of the unreacted GMP. At reaction times of 16 h and 32 h, the unreacted GMP band has nearly disappeared and the glycated GMP band represented by the smeared band has grown substantially more intense and shifted up in molecular mass.

SDS-PAGE gel was stained by SYPRO Red (Figure 6). Unreacted GMP with its characteristic king’s crown shape gradually disappeared over 32 h of reaction time. At the same time the glycated GMP smear band grew in intensity and shifted up to a higher molecular mass.

Figure 7 contains the result of fluorescence laser densitometry of the SYPRO Red stained gel of Figure 6. At 8 h of heating time, ε = 0.66%, or 66% of the un-glycated GMP had disappeared from the sample. At 32 h, ε = 0.87, or 87% of the un-glycated GMP had disappeared. The chromatographic procedure did not work for GMP because of its low isoelectric point of 3.15 to 4.15 [26]. The data for un-glycated GMP were fit using the model of Equation (2). The fitted parameter values were k = 0.13 ± 0.02 h^−1^ and [P]_eq_ = 1.50 ± 0.08 g/L. There was no statistical difference between the data points and the fit using the kinetic model (*p* > 0.05). Dextran glycation of GMP was substantially slower than dextran glycation of WPI.

## 4. Discussion

This work examined the kinetics and yield of whey protein glycation using dextran and the dry-heating method. Previous researchers have centered focus on the food properties of the un-purified reaction mixture after heating, but the kinetics of the protein-polysaccharide glycation reaction has been largely unstudied. The appropriate reaction time at 70 °C can be assessed by the reaction half-life (t_1/2_). From the disappearance of un-glycated protein, glycation of WPI had t_1/2_ = 1.8 h for WPI as measured by the chromatographic method, and t_1/2_ = 2.1 h for WPI as measured by the SDS-PAGE method. These times are not greatly different than the work of Akhtar and Dickinson [14] where a heating time of 2 h at 80 °C was used for the dry-heating method to glycate WPI using maltodextrin.

Glycation of GMP had t_1/2_ = 5.3 h at 70 °C as measured by the SDS-PAGE method. Conversion at 32 h was 87% for GMP. Glycation of GMP by dextran was substantially (61%) slower than glycation of WPI, although yield was similar to WPI. GMP when pure has an unusual amino acid composition compared to WPI. GMP is deficient in Asp, Leu, and Lys, compared to WPI, and is missing six amino acids altogether: Cys, Tyr, Phe, Trp, His, and Arg [25]. Furthermore, GMP contains double the amount of the following five amino acids compared to WPI: Thr, Ile, Pro, Ser, Val. Considering that Schiff base formation involves primarily Lys, and to a lesser extent His and Trp [9], and that GMP has half the Lys content of WPI, and no His and Trp, then it is logical that glycation of GMP was slower than for WPI.

The WPI-dextran and the GMP-dextran glycation reactions were well described by the kinetic model. The reason the error bars were larger for Figure 7 than Figure 4 was the greater uncertainty in calculating the band area for the irregular king’s crown shape of Figure 6 versus the regular rectangular shape of Figure 2. For WPI-dextran, there were no statistical differences between the data points and the fit using the kinetic model (*p* > 0.05) for 20 out of 21 time points. For GMP-dextran, there were no statistical differences between the data points and the fit using the kinetic model for seven out of seven time points (*p* > 0.05). These observations support the hypothesis that protein-dextran glycation using the dry-heating method is an apparent first-order reversible reaction.

For the WPI-dextran reaction, the rate constants measured for the disappearance of un-glycated protein by SDS-PAGE (k = 0.33 h^−1^) and by chromatographic analysis (k = 0.38 h^−1^) were not statistically different (*p* > 0.05). The rate constant for creation of glycated protein by chromatographic analysis (k = 0.18 h^−1^) was about half the expected value. This discrepancy in the rate constant for creation of glycated protein was because of the last two data points of Figure 4 (16 and 32 h) where the peak area for glycated protein [PD] continued to rise, while the peak area for un-glycated protein [P] did not continue to fall. It is possible that the protein extinction coefficient increases with glycation causing the peak area at 280 nm for un-glycated and glycated protein to be different, especially at long times where the extent of glycation is highest.

Yield was not previously measured for the dry-heating method. In the present work, yield for the glycation of WPI by dextran at 8 h was 88% for un-glycated protein based on chromatographic analysis, and 89% based on SDS-PAGE analysis. For comparison, the wet-heating method for dextran and WPI had at best a yield of 18% using a reaction time of 24 h at 60 °C [18]. In summary, the dry-heating method increased yield from 18% to nearly 90% and cut the reaction time from 24 h to 8 h for dextran and WPI glycation. These results are not surprising if one considers that the glycation reaction is a reversible condensation reaction. Condensation reactions eliminate water and should be run in a dry state to achieve high yield.

## 5. Conclusions

This work examined the kinetics of the dry-heating method for the glycation of WPI and GMP by dextran. The dry-heating method had a reaction half-life of about 2 h at 70 °C and about 90% yield. The dry-heating method had a 5× higher yield than the wet-heating method. Glycation using the dry-heating method was well described using a reversible first-order kinetic model. As the reaction time increased, the amount of un-glycated protein fell exponentially, and the amount of glycated protein rose exponentially. Glycation of GMP was slower than glycation of WPI. The half-life for GMP glycation was about 5 h compared to 2 h for WPI glycation. This was attributed to the lower Lys and other reactive amino acid content of GMP compared to WPI. The glycation reaction starts with Schiff base formation which is a reversible condensation reaction that makes water. By Le Chatelier’s principle, the presence of water in the reaction mixture drives the reaction in reverse, lowering yield. This may explain why the dry-heating method of protein glycation by dextran was superior to the wet-heating method in terms of speed and yield. It is important to note that reversible condensation reactions can be driven in reverse (towards reactants in Equation (1)) by adding water. The reverse reaction of Equation (1) happens when protein-dextran glycates in powder form are added to wet foods such as beverages, emulsions, gels, and foams. In this case the glycates may fall apart by hydrolysis and return to the form of the reactants: free un-glycated protein and dextran. Glycation of proteins by dextran via the Maillard reaction is a food-grade method to improve the physical properties of proteins for expanded use of proteins in foods. The present work is useful to the food industry to expand the use of glycated proteins in creating new food products.

## Figures and Tables

**Figure 1 foods-08-00528-f001:**
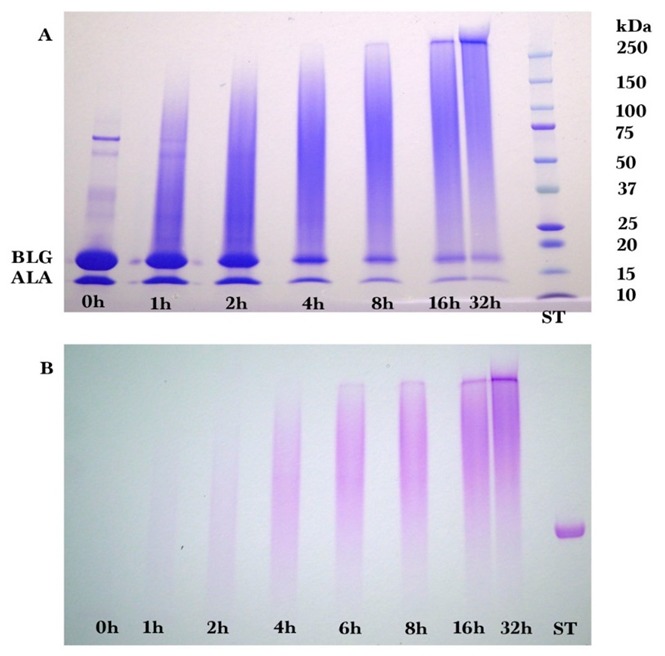
SDS-PAGE result for whey protein isolate-dextran glycation after (**A**) Coomassie Blue and (**B**) glycoprotein staining for reaction times of 0, 1, 2, 4, 8, 16, and 32 h at 70 °C. ST = protein standards, ALA = alpha-lactalbumin, BLG = beta-lactoglobulin, SDS-PAGE = sodium dodecyl sulfate-polyacrylamide.

**Figure 2 foods-08-00528-f002:**
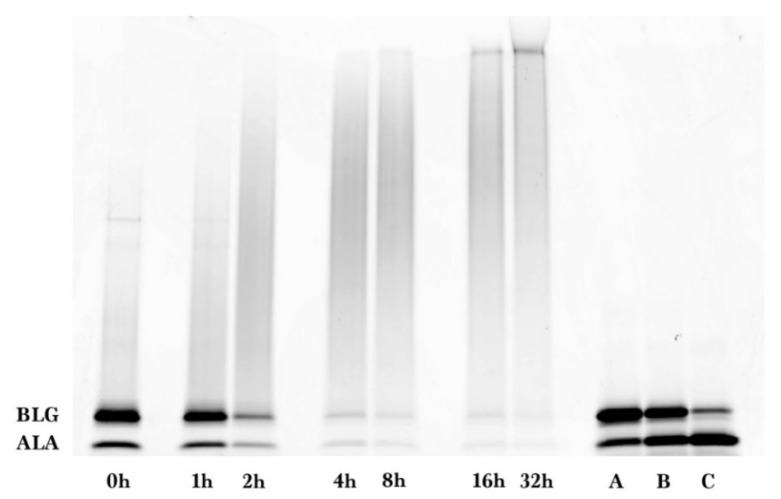
SDS-PAGE gel for whey protein isolate-dextran glycation after SYPRO Red staining of reaction mixtures made at the indicated heating times, and of internal standards (lanes A, B, C).

**Figure 3 foods-08-00528-f003:**
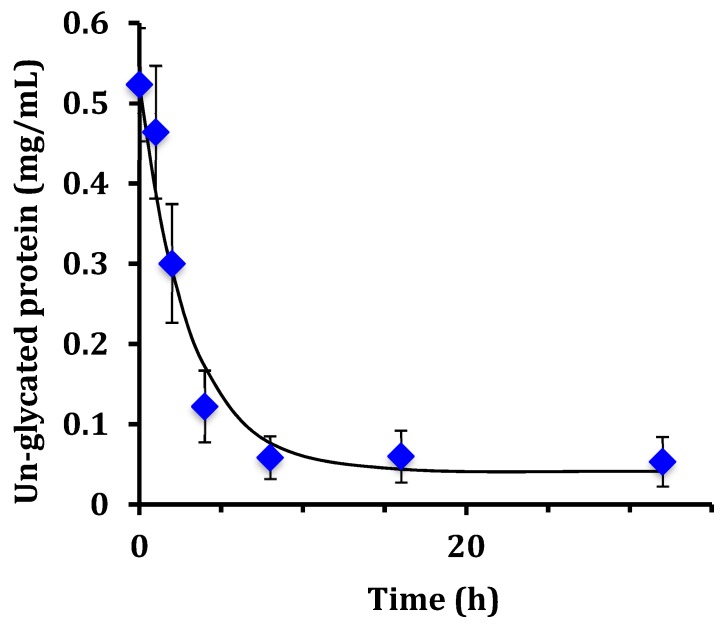
Un-glycated WPI versus reaction time at 70 °C determined by laser fluorescence densitometry (diamond markers) and the fit using the kinetic model of Equation (2) (solid line). Error bars are ± standard deviation of triplicates of the entire procedure of Section 2.1.

**Figure 4 foods-08-00528-f004:**
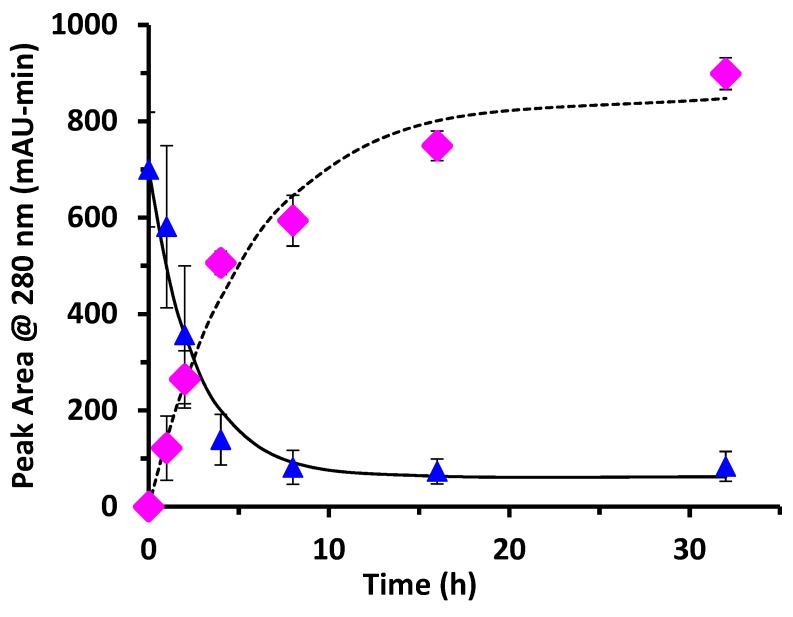
WPI-dextran glycation reaction at 70 °C determined by chromatographic analysis showing the loss of un-glycated protein (triangles) and simultaneous formation of glycated protein (diamonds). Un-glycated and glycated protein share the same Y-axis units (mAU-min). Also shown are the fit of the data using the kinetic model of Equation (2) for un-glycated protein (solid line) and Equation (3) for glycated protein (dotted line). Error bars are ± standard deviation of triplicates of the entire procedure of Section 2.1. WPI = whey protein isolate.

**Figure 5 foods-08-00528-f005:**
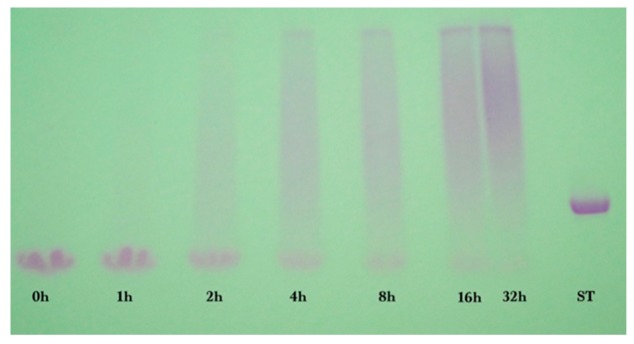
SDS-PAGE gel after glycoprotein staining, showing the appearance of glycated GMP as reaction time at 70 °C increases. GMP = glycomacropeptide

**Figure 6 foods-08-00528-f006:**
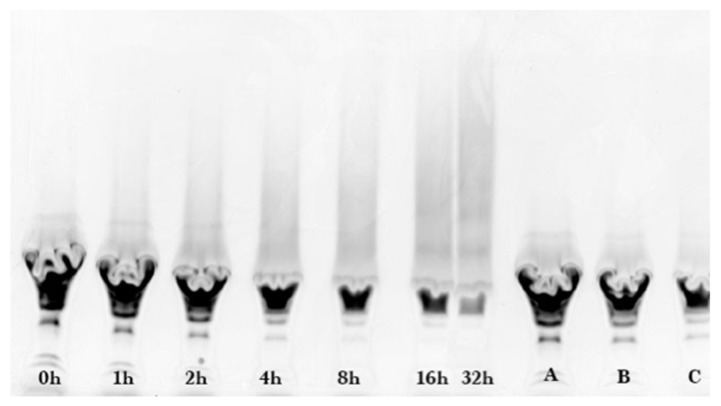
SDS-PAGE gel after SYPRO Red staining, showing the disappearance of un-glycated GMP over time. Lanes A, B, and C are calibration standards of known GMP concentration.

**Figure 7 foods-08-00528-f007:**
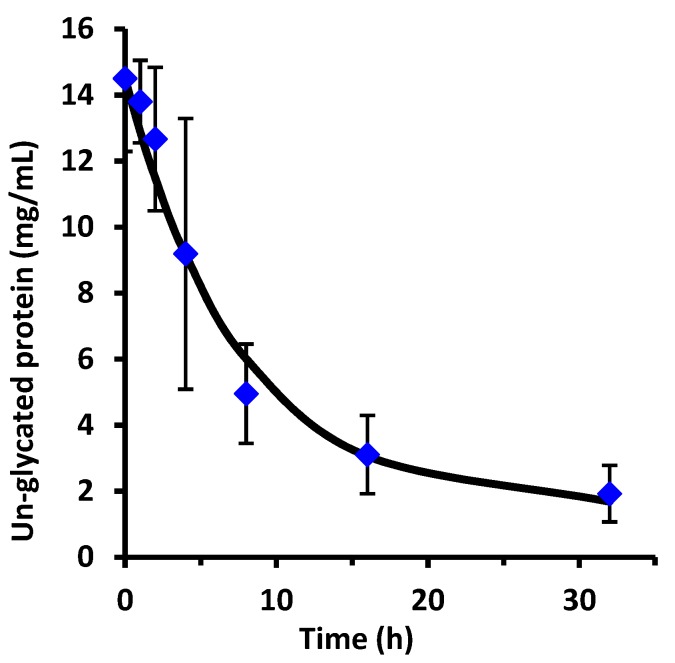
Un-glycated protein (GMP) versus reaction time at 70 °C determined by laser fluorescence densitometry (diamond markers) and fit using the kinetic model of Equation (2) (solid line). Error bars are ± standard deviation of triplicate experiments.

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
