# Peer review of "Kinetics of Whey Protein Glycation Using Dextran and the Dry-Heating Method"

_foods, 2019, doi:10.3390/foods8110528_

Round 1

Reviewer 1 Report

Manuscript entitled: “Kinetics of whey protein glycation using dextran and 2 the dry-heating method”.

General considerations:

The manuscript is interesting and within the scope of Foods journal. The English in several sections is colloquial and must be revised to an acceptable scientific format, and should be polished. In my opinion, the main issue of this manuscript is the lack of clarity. The authors should perform a serious effort to improve the clarity, particularly on the description of the statistical analysis performed. Information presented in the figures should be improved in terms of scientific quality.  

Point-by-point comments:

Celsius degrees and percentage are units, therefore must be separated from their numerical value. Please revise throughout the manuscript.

Line 22, SDS-PAGE is not properly defined.

Line 33, “Maillard reaction is a promising new food-grade method to modify food protein” please define: new.

Line 42, “minutes to days” is in my opinion highly vague. In my understanding, the impact of this section would be increased if the authors defined the minimum and maximum time values

Line 44, considering that Maillard reaction is a central reaction of this manuscript, please consider its thorough description, by providing the containing the materials used in the manuscript.

Line 74, what were the conditions used for dextran and glycomacropeptide?

Line 76, “ground into a powder” can the authors please further describe the grounding process? What were the mean size of the obtained particles?

Line 90 to 92, “Coomassie stained gels were useful to contrast the difference between un-glycated and glycated proteins but was not sufficiently quantitative because of poor contrast between the gel background and the protein bands” in my opinion this belongs in the Discussion section.

Line 112, the equations should be referenced in text, correct?

Line 115, please revise the colloquial language.

Line 117, please consider replacing “at any time t” to “at time t”

Figure 1 B, to what correspond the single ST band? Can the authors please include that information? And more importantly, is the information comprised of this figure corresponding to dextran and whey protein isolate, or dextran and glycomacropeptide? In my opinion this is not clear in the figure caption.

Line 128, “shifted up as reaction time increased” please revise English.

Line 128, “By 32 h of reaction time, little un-glycated ALA and BLG remained, and some of the glycated protein did not enter the gel” can the authors please clarify this sentence? I simply could not understand this statement.

Figure 2, the internal standards should be described in the Materials and Methods section.

Line 146, WPI was already previously properly defined.

Line 147, which equation is equation 1.2?

Line 148, “There was no statistical 148 difference between the data points and the fit using the kinetic model (p > 0.05)” how did the authors compare the results and the model? Which statistical analysis was performed? In my opinion this information should be displayed in a “Statistic analysis” section in the Materials and Methods section.

Figure 3, in my opinion, Y-axis caption is incomplete. Please revise.

Line 155, “triplicate experiments” correspond to 3 different glycations and each reaction product in 1 SDS-PAGE gel? Please clarify.

Line 159, I beg your pardon, but I do not clearly understand, SYPRO Red staining also has a smear, how was it quantified to achieve the results in Figure 3? Please clarify.

Figure 4 must be revised. Do the triangles correspond to peak area? In addition, the authors should perform a chromatographic calibration curve, and accurately convert peak area to concentration.

Line 189, “characteristic “king’s crown” shape of unreacted GMP” can the authors please provide a reference to support this statement?

No chromatographic analysis was performed for GMP?

Line 222, improperly formatted equation, without any in-text references.

Line 231, “substantially more of the following 5 amino acids” how much more?

The authors solely tested one temperature?

Line 265, please revise colloquial English.

Line 271, please clarify the following statement: “It is important to note that reversible condensation reactions can be driven in reverse by adding water”.

Line 272, please revise colloquial language.

Line 276, “improve the physical properties of proteins” no analysis of the physical properties was performed, correct?

Author Response

Response to Reviewer 1

General considerations:
The manuscript is interesting and within the scope of Foods journal. The English in several sections is colloquial and must be revised to an acceptable scientific format, and should be polished. In my opinion, the main issue of this manuscript is the lack of clarity. The authors should perform a serious effort to improve the clarity, particularly on the description of the statistical analysis performed. Information presented in the figures should be improved in terms of scientific quality.

Point-by-point comments:
Celsius degrees and percentage are units, therefore must be separated from their numerical value. Please revise throughout the manuscript.

As suggested, a space was added between the number and °C or % in each occurrence.

Line 22, SDS-PAGE is not properly defined.

As suggested, the spelled-out definition of SDS-PAGE was added to L22-23 in the Track Changes version.

Line 33, “Maillard reaction is a promising new food-grade method to modify food protein” please define: new.

The word “new” was deleted in L34 of the Track Changes version.

Line 42, “minutes to days” is in my opinion highly vague. In my understanding, the impact of this section would be increased if the authors defined the minimum and maximum time values

As suggested, numerical values were substituted for the words “minutes to days” in L42-44 of the Track Changes version.

Line 44, considering that Maillard reaction is a central reaction of this manuscript, please consider its thorough description, by providing the containing the materials used in the manuscript.

As suggested, a more detailed description of the Maillard reaction was added to the revised version of the manuscript in L68-81 of the Track Changes version.

Line 74, what were the conditions used for dextran and glycomacropeptide?

In L104 of the Track Changes version, the conditions for GMP were noted to be the same as for WPI.

Line 76, “ground into a powder” can the authors please further describe the grounding process? What were the mean size of the obtained particles?

As suggested, the description of the grinding process and the size of the particles was added to the revised manuscript in L107 of the Track Changes version.

Line 90 to 92, “Coomassie stained gels were useful to contrast the difference between un-glycated and glycated proteins but was not sufficiently quantitative because of poor contrast between the gel background and the protein bands” in my opinion this belongs in the Discussion section.

This section was revised to clarify in L157-158 of the Track Changes version.

Line 112, the equations should be referenced in text, correct?

As suggested, all equations were referenced in the text of the revised manuscript.

Line 115, please revise the colloquial language.

As suggested, the word “great” was replaced with more specific language in L187-188 of the Track Changes version.

Line 117, please consider replacing “at any time t” to “at time t”

As suggested, this change was made in L217 of the Track Changes version.

Figure 1 B, to what correspond the single ST band? Can the authors please include that information? And more importantly, is the information comprised of this figure corresponding to dextran and whey protein isolate, or dextran and glycomacropeptide? In my opinion this is not clear in the figure caption.

As suggested, the description of the ST band in Figure 1 was added to L152-156 of the Track Changes version, and the caption for Figure 1 was changed to indicate that it refers to whey protein isolate and dextran.

Line 128, “shifted up as reaction time increased” please revise English.

As suggested, this sentence was revised to clarify in L236-238 of the Track Changes version.

Line 128, “By 32 h of reaction time, little un-glycated ALA and BLG remained, and some of the glycated protein did not enter the gel” can the authors please clarify this sentence? I simply could not understand this statement.

As suggested, this sentence was changed to clarify in L238-240 of the Track Changes version.

Figure 2, the internal standards should be described in the Materials and Methods section.

As suggested, the internal standards were described in L162-165 of the Track Changes version.

Line 146, WPI was already previously properly defined.

As suggested, the unnecessary duplicate definition of WPI was eliminated from the revised manuscript in L264 of the Track Changes version.

Line 147, which equation is equation 1.2?

Equation 1.2 is the equation for [P], the un-glycated protein in L190 of the Track Changes version.

Line 148, “There was no statistical 148 difference between the data points and the fit using the kinetic model (p > 0.05)” how did the authors compare the results and the model? Which statistical analysis was performed? In my opinion this information should be displayed in a “Statistic analysis” section in the Materials and Methods section.

As suggested, a “statistical analysis” section was added to the Materials and Methods section of the revised manuscript in L221-226 of the Track Changes version.

Figure 3, in my opinion, Y-axis caption is incomplete. Please revise.

As suggested, the Y-axis caption for Figure 3 was revised.

Line 155, “triplicate experiments” correspond to 3 different glycations and each reaction product in 1 SDS-PAGE gel? Please clarify.

As suggested, the meaning of “triplicate” was clarified in the caption for Figure 3.

Line 159, I beg your pardon, but I do not clearly understand, SYPRO Red staining also has a smear, how was it quantified to achieve the results in Figure 3? Please clarify.

Thank you for this point, because we realized there was a confusing typo. We revised the manuscript to correct this error in L290-291 of the Track Changes version.

Figure 4 must be revised. Do the triangles correspond to peak area? In addition, the authors should perform a chromatographic calibration curve, and accurately convert peak area to concentration.

Yes, the triangles do refer to peak area. As suggested, the calibration between peak area and protein was added to L180-181 of the Track Changes version.

Line 189, “characteristic “king’s crown” shape of unreacted GMP” can the authors please provide a reference to support this statement?

As suggested, a reference was added to support the “king’s crown” shape of unreacted GMP in L322 of the Track Changes version.

No chromatographic analysis was performed for GMP?

No, it was not as explained in L 353-354 of the Track Changes version.

Line 222, improperly formatted equation, without any in-text references.

This equation was deleted from the in-text area. The definition is not needed as it appears in essentially all general chemistry textbooks.

Line 231, “substantially more of the following 5 amino acids” how much more?

As suggested, a specific number was calculated and added to L386-387 of the Track Changes version.

The authors solely tested one temperature?

That is correct. Only 70 °C and 80 % relative humidity were tested as stated in the Abstract.

Line 265, please revise colloquial English.

As suggested, the sentence was revised in L438-440 of the Track Changes version.

Line 271, please clarify the following statement: “It is important to note that reversible condensation reactions can be driven in reverse by adding water”.

As suggested, the sentence was revised in L438-440 of the Track Changes version.

Line 272, please revise colloquial language.

This sentence was deleted.

Line 276, “improve the physical properties of proteins” no analysis of the physical properties was performed, correct?

That is correct.

Reviewer 2 Report

The paper is characterized by several limitations listed in the following bulleted points

Glycation of protein is not a new food grade technology to modify protein structure. First mass spectromety studies on the modification of protein and on the significance of the molecules formed have been performed around 30 years ago. See the following: Ledl, Franz, and Erwin Schleicher. "New aspects of the Maillard reaction in foods and in the human body." Angewandte Chemie International Edition in English 29.6 (1990): 565-594.

Schiff base is not a stable compound and this sugar-amino compounds is immediately converted into the more stable Amadori ( in the case of aldhoses) or Heyns compounds (in the case of reaction with ketoses). It would be worthy to mention the relevance of the first stable compounds and which analytical techniques can provide information on the extent of glycation.

Line 52: remove that sentence. What is the meaning of kinetic? Are the authors interested in the chemical background of compound formation/reactants disappearance or only a fitting procedure? This is a crucial aspect that need to be detailed in the introduction section (Martins, Sara IFS, Wim MF Jongen, and Martinus AJS Van Boekel. "A review of Maillard reaction in food and implications to kinetic modelling." Trends in food science & technology 11.9-10 (2000): 364-373.)  

Did the authors take into consideration 1% lactose present in GMP?

Quantitative relevance and analytically exact characterization of compounds/precursors is mandatory to build a solid kinetic profile (Hellwig, Michael, et al. "Quality criteria for studies on dietary glycation compounds and human health." Journal of agricultural and food chemistry (2019)). In this respect validation of the analytical procedure is missing and further information are mandatory. The authors used a cation exchange chromatography with UV detection at 280 nm, but analytical performance cannot be confined to the elution profile and further characterizations are mandatory, in particular for the significance of the molecules formed. The following technical report highlights how many efforts are necessary to characterize the compounds formed in this kind of experiments. Pure Appl. Chem. 2019; 91(8): 1417–1437

Line 112: the reaction route cannot be stopped at the formation of Schiff base. The authors should be aware that behind formation of Schiff base other reactions are possible. The first one is the formation of the Amadori compounds the second one is the fragmentation and autoxidation of Amadori moiety that yield intermediates and end products. In the frame of the peeling-off mechanisms (see Kroh and coworkers) free reducing sugar moieties are released and they can contribute to the glycation.

Which was the effect of lactose? Furosine measurement can provide information on the total glycation and on the difference between whey and GMP.

Figure 4: how the authors are sure that there is no overlapping of the target analytes with other interferences? Moreover, as no differential equation was built and the residuals of the model proposed were not studied, the procedure can be associated to a fitting procedure without any answer to the kinetic model proposed. 

Figure 7: how did the authors explain the huge variation of the different time points?

Line 218: here it is a partial and incomplete list of some of the references that deal with kinetic of protein glycation and Maillard reaction: 

Ajandouz, El Hassan, et al. "Effects of temperature and pH on the kinetics of caramelisation, protein cross-linking and Maillard reactions in aqueous model systems." Food Chemistry 107.3 (2008): 1244-1252.

Claeys, W. L., Ludikhuyze, L. R., & HENDRICKX, M. E. (2001). Formation kinetics of hydroxymethylfurfural, lactulose and furosine in milk heated under isothermal and non-isothermal conditions. Journal of Dairy Research68(2), 287-301.

Aalaei, Kataneh, Marilyn Rayner, and Ingegerd Sjöholm. "Kinetics of available lysine in stored commercial skim milk powder at moderate temperatures." International journal of food science & technology 53.9 (2018): 2159-2165. 

Author Response

Response to Reviewer 2

The paper is characterized by several limitations listed in the following bulleted points

Glycation of protein is not a new food grade technology to modify protein structure. First mass spectromety studies on the modification of protein and on the significance of the molecules formed have been performed around 30 years ago. See the following: Ledl, Franz, and Erwin Schleicher. "New aspects of the Maillard reaction in foods and in the human body." Angewandte Chemie International Edition in English 29.6 (1990): 565-594.

We agree with this point and it helped us to realize an unwritten assumption we made in the manuscript. The Maillard reaction is commonly present in foods as the browning reaction between a reducing sugar and a protein. We failed to point out that we are not looking at this reaction. We have added a lengthy section to the revised manuscript to correct our mistake. Please see L68-81 of the Track Changes version.

Schiff base is not a stable compound and this sugar-amino compounds is immediately converted into the more stable Amadori ( in the case of aldhoses) or Heyns compounds (in the case of reaction with ketoses). It would be worthy to mention the relevance of the first stable compounds and which analytical techniques can provide information on the extent of glycation.

We agree with this statement. Because of our previously mentioned error, we can especially see the reason for this comment by the reviewer. We have made the revision to explain that our Maillard reaction is different from the Maillard browning reaction. Please see L68-81 of the Track Changes version.

Line 52: remove that sentence. What is the meaning of kinetic? Are the authors interested in the chemical background of compound formation/reactants disappearance or only a fitting procedure? This is a crucial aspect that need to be detailed in the introduction section (Martins, Sara IFS, Wim MF Jongen, and Martinus AJS Van Boekel. "A review of Maillard reaction in food and implications to kinetic modelling." Trends in food science & technology 11.9-10 (2000): 364-373.)

We agree with this comment. The word “kinetic” means something different to a chemist in that it often refers to defining the kinetic pathway and series of elementary reactions that occur in that pathway. We use the word “kinetic” to mean simply the time course of a reaction. We changed to text to reflect this distinction in L68-81 of the Track Changes version.

Did the authors take into consideration 1% lactose present in GMP?

Yes, we considered dialysis to remove the 1% lactose from the glycomacropeptide. This is a good point. Our past research showed that dialysis to remove traces of simple residual sugars does not affect the reaction between the protein and the polysaccharide. Other researchers have found dialysis of simple sugars out of the protein sample is unnecessary. We have added note of these points to the revised manuscript in L93-94 of the Track Changes version.

Quantitative relevance and analytically exact characterization of compounds/precursors is mandatory to build a solid kinetic profile (Hellwig, Michael, et al. "Quality criteria for studies on dietary glycation compounds and human health." Journal of agricultural and food chemistry (2019)). In this respect validation of the analytical procedure is missing and further information are mandatory. The authors used a cation exchange chromatography with UV detection at 280 nm, but analytical performance cannot be confined to the elution profile and further characterizations are mandatory, in particular for the significance of the molecules formed. The following technical report highlights how many efforts are necessary to characterize the compounds formed in this kind of experiments. Pure Appl. Chem. 2019; 91(8): 1417–1437

We agree with this point. As mentioned above, we failed to clarify the distinction between the Maillard browning reaction pathway and our work. In addition, we used the word “kinetics” which means something different to a chemist that studies reaction pathways than to us when simply following the time course of a reaction. We have tried to remedy the situation by adding the lengthy section mentioned above to the revised manuscript to correct our mistake. Please see L68-81 of the Track Changes version.

Line 112: the reaction route cannot be stopped at the formation of Schiff base. The authors should be aware that behind formation of Schiff base other reactions are possible. The first one is the formation of the Amadori compounds the second one is the fragmentation and autoxidation of Amadori moiety that yield intermediates and end products. In the frame of the peeling-off mechanisms (see Kroh and coworkers) free reducing sugar moieties are released and they can contribute to the glycation.

We agree with this point and admit that we inadvertently created this confusion by failing to clarify the distinction between the Maillard browning reaction pathway and our work. Please see L68-81 of the Track Changes version where we have tried to clarify the situation.

Which was the effect of lactose? Furosine measurement can provide information on the total glycation and on the difference between whey and GMP.

Please see our response to the 1% lactose question above.

Figure 4: how the authors are sure that there is no overlapping of the target analytes with other interferences? Moreover, as no differential equation was built and the residuals of the model proposed were not studied, the procedure can be associated to a fitting procedure without any answer to the kinetic model proposed.

We agree with this comment. We did not build a kinetic model in the sense of discovery a chemical pathway and the intermediates in that pathway. We agree that ours is basically a fitting procedure of a simple model to follow the time course of the data for the reaction between the protein and the polysaccharide. We have now stated this outright in the revised manuscript in L77-81 of the Track Changes version.

Figure 7: how did the authors explain the huge variation of the different time points?

We added the reason for the larger variation in the GMP versus WPI data in L392-394 of the Track Changes version. Please note also that we were unable to use the chromatographic method for GMP for the reasons given in L353-354 of the Track Changes version.

Line 218: here it is a partial and incomplete list of some of the references that deal with kinetic of protein glycation and Maillard reaction:
Ajandouz, El Hassan, et al. "Effects of temperature and pH on the kinetics of caramelisation, protein cross-linking and Maillard reactions in aqueous model systems." Food Chemistry 107.3 (2008): 1244-1252.
Claeys, W. L., Ludikhuyze, L. R., & HENDRICKX, M. E. (2001). Formation kinetics of hydroxymethylfurfural, lactulose and furosine in milk heated under isothermal and non-isothermal conditions. Journal of Dairy Research, 68(2), 287-301.
Aalaei, Kataneh, Marilyn Rayner, and Ingegerd Sjöholm. "Kinetics of available lysine in stored commercial skim milk powder at moderate temperatures." International journal of food science & technology 53.9 (2018): 2159-2165.

Round 2

Reviewer 1 Report

The manuscript entitled: “Kinetics of whey protein glycation using dextran and the dry-heating method”, reference: foods-605438

I would like to congratulate the authors for their efforts in improving the clarity, and impact of the manuscript.

I solely suggest a single minor amendment:

In Figure 4, can authors convert peak area to protein concentration mg/mL?  In addition, Figure 4 should have a second Y-axis displaying glycated protein (diamonds) and its units.

Author Response

I would like to congratulate the authors for their efforts in improving the clarity, and impact of the manuscript.

I solely suggest a single minor amendment:

In Figure 4, can authors convert peak area to protein concentration mg/mL? In addition, Figure 4 should have a second Y-axis displaying glycated protein (diamonds) and its units.

The peak area in units of mAU-min was the raw data collected in the laboratory. To convert those units to concentration would require making several assumptions, one of which is that the extinction coefficient of the protein is constant as the degree of glycation changes from zero to completion. As mentioned in L451-456 of the revised manuscript, this assumption is suspect. Therefore, we cannot justify changing the units of the y-axis in Figure 4.

We added a sentence to the caption for Figure 4 stating that the data for glycated protein (diamonds) share the same units as the data for un-glycated protein (triangles). This is why a second axis for glycated protein is not necessary.